# Angiotensin IV Receptors in the Rat Prefrontal Cortex: Neuronal Expression and NMDA Inhibition

**DOI:** 10.3390/biomedicines13010071

**Published:** 2024-12-31

**Authors:** Zsolt Tamás Papp, Polett Ribiczey, Erzsébet Kató, Zsuzsanna E. Tóth, Zoltán V. Varga, Zoltán Giricz, Adrienn Hanuska, Mahmoud Al-Khrasani, Ákos Zsembery, Tibor Zelles, Laszlo G. Harsing, László Köles

**Affiliations:** 1Department of Oral Biology, Semmelweis University, H-1089 Budapest, Hungary; papp.zsolt@semmelweis.hu (Z.T.P.); ribiczey.polett@gmail.com (P.R.); hanuska.adrienn@dpckorhaz.hu (A.H.); zsembery.akos@semmelweis.hu (Á.Z.); zelles.tibor@semmelweis.hu (T.Z.); 2Department of Pharmacology and Pharmacotherapy, Semmelweis University, H-1089 Budapest, Hungary; kato.erzsebet@semmelweis.hu (E.K.); varga.zoltan@semmelweis.hu (Z.V.V.); giricz.zoltan@semmelweis.hu (Z.G.); al-khrasani.mahmoud@semmelweis.hu (M.A.-K.); harsing.laszlo@semmelweis.hu (L.G.H.J.); 3Laboratory of Neuroendocrinology and In Situ Hybridization, Department of Anatomy, Histology and Embryology, Semmelweis University, H-1094 Budapest, Hungary; toth.zsuzsanna.emese@semmelweis.hu; 4Laboratory of Molecular Pharmacology, HUN-REN Institute of Experimental Medicine, H-1083 Budapest, Hungary

**Keywords:** N-methyl-D-aspartate receptor, AT_4_ angiotensin receptor, insulin-regulated aminopeptidase, prefrontal cortex, renin–angiotensin system, neuromodulation

## Abstract

Background: N-methyl-D-aspartate type glutamate receptors (NMDARs) are fundamental to neuronal physiology and pathophysiology. The prefrontal cortex (PFC), a key region for cognitive function, is heavily implicated in neuropsychiatric disorders, positioning the modulation of its glutamatergic neurotransmission as a promising therapeutic target. Our recently published findings indicate that AT_1_ receptor activation enhances NMDAR activity in layer V pyramidal neurons of the rat PFC. At the same time, it suggests that alternative angiotensin pathways, presumably involving AT_4_ receptors (AT4Rs), might exert inhibitory effects. Angiotensin IV (Ang IV) and its analogs have demonstrated cognitive benefits in animal models of learning and memory deficits. Methods: Immunohistochemistry and whole-cell patch-clamp techniques were used to map the cell-type-specific localization of AT4R, identical to insulin-regulated aminopeptidase (IRAP), and to investigate the modulatory effects of Ang IV on NMDAR function in layer V pyramidal cells of the rat PFC. Results: AT4R/IRAP expression was detected in pyramidal cells and GABAergic interneurons, but not in microglia or astrocytes, in layer V of the PFC in 9–12-day-old and 6-month-old rats. NMDA (30 μM) induced stable inward cation currents, significantly inhibited by Ang IV (1 nM–1 µM) in a subset of pyramidal neurons. This inhibition was reproduced by the IRAP inhibitor LVVYP-H7 (10–100 nM). Synaptic isolation of pyramidal neurons did not affect the Ang IV-mediated inhibition of NMDA currents. Conclusions: Ang IV/IRAP-mediated inhibition of NMDA currents in layer V pyramidal neurons of the PFC may represent a way of regulating cognitive functions and thus a potential pharmacological target for cognitive impairments and related neuropsychiatric disorders.

## 1. Introduction

The prefrontal cortex (PFC) is localized at the anterior pole of the mammalian brain. Its cytoarchitectural structure consists of a six-layered isocortex. The orbitomedial region is an agranular cortex distinguished by prominent layers V and VI and the absence of layer IV. The PFC is highly interconnected, receiving projections and sending fibers to numerous brain structures. It contains a wide range of well-known neurotransmitters, including glutamate, as well as more recently recognized neuromodulators such as angiotensin peptides. The PFC plays essential roles in planning, decision-making, and executive attention. It is one of the first brain regions to show changes during normal aging and is also among the most vulnerable to neurodegenerative alterations in dementia [1,2].

Glutamate is the primary excitatory neurotransmitter in the nervous system, and the N-methyl-D-aspartate receptor (NMDAR) is one of the most prominent of its receptors. In addition to agonist binding, NMDAR activation requires a co-agonist, glycine [3], and exhibits voltage-dependent sensitivity to magnesium [4]. NMDAR is a cation-selective ion channel with high permeability to calcium ions. While it can produce a short-term depolarizing effect, its primary, more sustained signaling mechanism involves the activation of the calcium/calmodulin-dependent protein kinase II (CaMKII) pathway [5]. Glutamate and NMDAR play crucial roles in physiologic functions such as learning and memory through activity-dependent synaptic plasticity [6]. However, their overstimulation is implicated in neurodegenerative disorders due to cytotoxic effects [7]. Memantine, a low-affinity, non-competitive NMDAR antagonist, is approved for the treatment of moderate to severe Alzheimer’s disease.

The classical axis of the renin–angiotensin system (RAS)—comprising angiotensinogen, renin, ACE, angiotensin II (Ang II), and the AT_1_ receptor (AT1R)—has been well established. Following the discovery of local RAS systems, including the brain RAS, subsequent research has identified new RAS axes in the brain, such as the angiotensin III (Ang III)/aminopeptidase N (APN)/Ang IV/IRAP/AT4R axis [8]. Ang IV is a hexapeptide derived from angiotensinogen that binds with high affinity to its unique receptor, AT4R, in a saturable, reversible, and specific manner [9]. Subsequently, AT4R was identified as the transmembrane enzyme IRAP, which is inhibited potently by its competitive ligand Ang IV [10,11]. The AT4R subtype is expressed in various brain regions, including the neocortex [12,13,14]. AT4Rs appear to be predominantly expressed in neurons, with limited evidence for expression in astrocytes and no reported expression in microglia or oligodendrocytes [8]. Ang IV and its analogs enhance long-term potentiation, support memory consolidation and retrieval, improve cognitive function, and mitigate memory deficits, as observed in animal models. Additionally, Ang IV promotes synaptic plasticity, exhibits antioxidant and anti-inflammatory effects, and provides neuroprotection in an ischemic stroke model [8,15].

Recently, we reported that Ang II potentiates NMDAR currents in a subset of layer V pyramidal cells in the rat PFC [16]. In the same study, we observed that higher concentrations of Ang II elicited inhibitory tendencies in a subgroup of pyramidal cells unresponsive to Ang II-induced potentiation. These effects were hypothesized to result from the action of Ang III and/or Ang IV formed by the conversion of Ang II into shorter angiotensin peptides. The present study aims to characterize the expression of AT4Rs in the predominant cell types of the rat PFC and to investigate the effects of Ang IV on the NMDAR current in layer V pyramidal cells. This investigation seeks to clarify the physiological relevance of these inhibitory responses and further elucidate the neuromodulatory role of angiotensin peptides in the PFC.

## 2. Materials and Methods

### 2.1. Experimental Animals

Nine-to-twelve-day-old, three-month-old, and six-month-old Wistar rats (Animal House of Semmelweis University, Hungary; Toxicoop, Hungary) were used in this study. All experiments were conducted in accordance with the guidelines and with the approval of the Ethical Board of Semmelweis University, Budapest, Hungary, and in compliance with the European Community’s Council Directives (2010/63/EU).

### 2.2. Electrophysiological Studies

#### 2.2.1. Brain Slice Preparation

Brain slices were prepared as previously described [17,18,19,20,21,22].

Wistar rat pups, aged 9–12 days, were decapitated, and their brains were rapidly extracted and placed in ice-cold, carbogenated (95% O_2_, 5% CO_2_) artificial cerebrospinal fluid (aCSF) with the following composition (mM): NaCl 126, KCl 2.5, NaH_2_PO_4_ 1.2, CaCl_2_ 2.4, MgCl_2_ 1.3, NaHCO_3_ 25, and glucose 11 (pH 7.4). Coronal slices (200 μm thick) containing the prelimbic region of the medial PFC (mPFC) were cut from a tissue block using a vibrating-blade microtome (MA752, Campden Instruments, Kensington, UK). The slices were then transferred to a holding chamber (Haake DC 10, Thermo Electron Corporation, USA; VWB2 2, VWR, England), where they were incubated in carbogenated aCSF at 36 °C for 40 min, followed by storage at room temperature (20–22 °C). For electrophysiological recordings, individual slices were placed in a recording chamber and continuously superfused with carbogenated aCSF at a flow rate of 3 mL/min using a peristaltic pump (MCP Standard, Ismatec, Germany) at room temperature (20–22 °C). Only one cell was measured per brain slice.

#### 2.2.2. Whole-Cell Patch-Clamp Recordings

Whole-cell patch-clamp recordings were performed using procedures similar to those described previously [17,18,19,20,21,22].

Pyramidal cells in layer V of the PFC were visualized using an upright microscope equipped with a ×40 water immersion objective (Axioscope FS; Carl Zeiss, Baden-Württemberg, Germany). Patch pipettes were fabricated using a vertical micropipette puller (PP-83, Narishige, Japan) from borosilicate glass capillaries. The pipettes were filled with a standard intracellular solution containing the following composition (mM): K-gluconate 140, NaCl 10, MgCl_2_ 1, HEPES 10, EGTA 11, Mg-ATP 1.5, and Li-GTP 0.3; pH 7.3, adjusted with KOH. The tip resistances of the pipettes ranged from 5 to 7 MΩ. Whole-cell access was routinely established and maintained for up to 40 min, with stable membrane properties throughout the recording period. First, the system was allowed to settle for 5 min to establish a diffusion equilibrium between the patch pipette and the cell interior. NMDA (30 μM) was then applied three times for 1.5 min (T_1_, T_2_, T_3_), with superfusion periods of 10 min using drug-free aCSF between each application. Membrane currents were recorded in the voltage-clamp configuration using an amplifier (Axopatch 200B, Molecular Devices, San Jose, CA, USA) at a holding potential of −70 mV. Suspected neuromodulators were present in the bath for 5 min prior to and during the third application of NMDA (T_3_). Their antagonists, as well as TTX (0.5 μM) or a Ca^2^⁺-free medium, were superfused throughout the experiment. Data were filtered at 2 kHz using the built-in low-pass filter of the amplifier, digitized at 5 kHz (Digidata 1200 series, Molecular Devices, San Jose, CA, USA), and stored on a laboratory computer.

#### 2.2.3. Data Analysis and Statistics

Data were analyzed using commercially available software (pClamp 10.2, Molecular Devices, San Jose, CA, USA). Reproducible inward currents were evoked in individual experiments; however, the amplitudes showed substantial variability across cells. Consequently, T_3_/T_2_ was expressed as the percentage change relative to the respective control response. Given the heterogeneity of the pyramidal neuron population and the variability in their responses, the following methodology was applied for response analysis. The exclusion criterion for inhibitory responses was defined based on the control mean, standard error of the mean (SEM), and the 99.9% confidence interval. The threshold, which depended on the parameters of the control experiments within the series, was typically around 10% inhibition. Neurons falling below this threshold were excluded from further analysis. Means ± SEM are presented throughout. Multiple comparisons with control values were performed using one-way ANOVA followed by Bonferroni’s post hoc test. A *p*-value of ≤0.05 was considered statistically significant.

### 2.3. Immunhistochemistry (IHC)

#### 2.3.1. Tissue Collection

Tissue collection was implemented based on as previously described [23]. Ten-day-old and six-month-old Wistar rats were anesthetized by inhalation of 5% isoflurane in a chamber and the intraperitoneal injection of ketamine (100 mg/kg) and xylazin (10 mg/kg). During surgery, animals were placed on crushed ice. After a bilateral thoracotomy, a tube connector (ISM580 or ISM583, Cole Palmer, Vernon Hills, IL, USA) was passed through the left ventricle into the ascending aorta. Through the beating heart, via the vascular system, 4% formalin, then PBS, was perfused by means of a peristaltic pump (MCP Standard, Ismatec, Germany) at room temperature (RT). Whole brains were carefully removed and placed in 4% formalin for 24 h at 4 °C. Following in vivo fixation, the brains were embedded in paraffin, dehydrated, and sliced at 4 μm.

#### 2.3.2. Immunocytochemistry

For the IHC, the deparaffinized and rehydrated sections underwent antigen retrieval in citrate buffer (pH 6.0) at 95 °C for 15 min, followed by washing three times for 5 min each in phosphate-buffered saline (PBS, pH 7.4) at RT. The inactivation of endogenous peroxidases was performed with 3% H_2_O_2_ treatment for 10 min at RT. After washing them three times for 5 min each in PBS, serum blocking was performed for 1 h at 4 °C with the following solution: PBS containing 2.5% bovine serum albumin, 2.5% non-fat dry milk, 2.5% horse serum, and 2.5% goat serum. For all IHC staining methods performed, primary antibodies were diluted in the complete blocking buffer used for serum blocking. At the end of all IHC staining, antibody labeling steps were followed by staining with 4′,6-diamidino-2-phenylindole dihydrochloride (DAPI; Thermo Fisher Scientific, Waltham, MA, USA), diluted 1:1000 in PBS, for 5 min at RT, followed by washing once for 10 min with gentle shaking in PBS. Finally, all slides were mounted with Fluoromount-G medium (Leica Biosystems, Nußloch, Germany).

In the case of 10-day-old and 6-month-old Wistar rats, the primary antibody, anti-IRAP (D2C5)XP^®^ rabbit mAb (Cell Signaling Technology, Danvers, MA, USA), was 100-times diluted, and sections were incubated for 1 h at RT. After washing steps in PBS containing 0.05% Tween, pH = 7.4, the secondary antibody SignalStain Boost IHC Detection Reagent (Cell Signaling Technology, Danvers, MA, USA) diluted 2 times in PBS containing 2.5% goat serum, was used for 45 min at RT. This was followed by washing five times for 10 min each, with gentle shaking in PBS containing 0.05% Tween. TSA-FITC (Akoya Biosciences, Marlborough, MA, USA) was diluted 1000 times in 1× Plus Amplification Diluent (Akoya Biosciences), and sections were incubated for 6 min at RT, followed by washing steps in PBS containing 0.05% Tween. At the end of the DAPI staining, washing steps and mounting were performed as described in the first section.

For the investigation of the cell-type-specific expression of IRAP/AT4R in 6-month-old Wistar rats, double staining was used as described in the previous section, and afterwards it was followed by staining with antibodies raised against cell-type-specific markers. For GABAergic interneurons, anti-GAD67 (1G10.2) mouse mAb (Sigma-Aldrich, St. Louis, MO, USA) was used; for astrocytes, anti-GFAP (G3893) mouse mAb (Sigma-Aldrich, St. Louis, MO, USA) was used; whereas for microglias, anti-Iba1 (GT10312) mouse mAb (Thermo Fisher Scientific, Waltham, MA, USA) was used. Anti-GAD67 (1G10.2) mouse mAb was 150-times diluted in complete blocker, whereas anti-GFAP (G3893) mouse mAb and anti-Iba1 (GT10312) mouse mAb were 100-times diluted. Sections were incubated with these antibodies overnight at 4 °C. The next day, after washing them 5 times for 10 min each with gentle shaking in PBS containing 0.05% Tween, sections were incubated in 500-times diluted anti-mouse IgG Fab2 Alexa Fluor^®^ 647 (Cell Signaling Technology, Danvers, MA, USA) for 1 h at RT. After washing them five times for 10 min each with gentle shaking with PBS containing 0.05% Tween DAPI stain, washing steps and mounting were performed as described in the first section. 

As a low expression of particular cell-type-specific markers was found in 10-day-old Wistar rats, a double IHC with tyramide signal amplification was used [24,25]. The first sections were stained for IRAP/AT4R the same way as described above. After incubation with TSA-FITC (Akoya Biosciences, Marlborough, MA, USA), followed by washing them five times for 10 min each with gentle shaking in PBS, a microwave treatment (MWT) was performed. Slides were placed in a black plastic container filled with 10 mM citric acid buffer, pH 6.0, and it was placed into an 800 W microwave oven. The liquid was then heated to the boiling point at 100% power in approximately 2.5 min. Afterwards, sections were further microwaved for 5 min at 50% power. The slides were allowed to cool down in citric acid buffer for 30 min at room temperature and then washed three times for 5 min each with PBS. After MWT, the primary antibodies against cell-type-specific markers (anti-GAD67 (1G10.2) mouse mAb (Sigma-Aldrich), anti-GFAP (G3893) mouse mAb (Sigma-Aldrich, St. Louis, MO, USA), and anti-Iba1 (GT10312) mouse mAb (Invitrogen)) were diluted 1:50, 1:100, and 1:100, respectively, and sections were incubated overnight at 4 °C. The next day, following washing them 5 times for 10 min each with gentle shaking in PBS containing 0.05% Tween, all samples were incubated with 1.7-times diluted ImmPRESS^®^ HRP horse-made, rat-adsorbed, anti-mouse IgG (Vector Laboratories, Newark, CA, USA) in PBS containing 2.5% goat serum for 1 h at RT. After washing them 5 times for 10 min each with gentle shaking with PBS containing 0.05% Tween, samples were incubated with 1000-times diluted TSA-Cy3 (Akoya Biosciences, Marlborough, MA, USA) in 1× Plus Amplification Diluent (Akoya Biosciences, Marlborough, MA, USA) for 6 min at RT. After subsequently washing them five times for 10 min each with gentle shaking in PBS containing 0.05% Tween, DAPI stain, the washing step and mounting were performed as described in the first section. 

## 3. Results

### 3.1. Ang II Inhibited NMDA-Induced Inward Currents in Layer V Pyramidal Neurons of the Rat PFC

Repetitive superfusion of NMDA (30 µM) in aCSF for 1.5 min (T_1_, T_2_, T_3_) with 10-min intervals between applications elicited reproducible inward cation currents during the second and third applications (T_2_ and T_3_), with a T_3_/T_2_ ratio of 98.22 ± 1.88% (Figure 1A, left column).

A repeated and more detailed analysis of the effects of a higher dose of Ang II on NMDA currents, as noted in our previous publication [16], revealed that the superfusion of Ang II (3 μM) for 5 min before and during T_3_ significantly inhibited NMDA-mediated responses (T_3_/T_2_ = 0.659 ± 0.037, corresponding to a −34.06% ± 3.77% change at T_3_; n = 9; *p* < 0.01), in a subset (28.1%) of neurons (Figure 1).

The inhibitory effect of Ang II was not abolished by AT_1_ receptor blockade. The application of the AT_1_ receptor antagonist eprosartan (1 μM) alone, throughout the measurement period, had no effect on any of the cells tested (T_3_/T_2_ = 0.989 ± 0.031, n = 7). In the presence of eprosartan, superfusion with Ang II (1 μM and 3 μM) inhibited NMDA-mediated responses in 30% of neurons (3 out of 10; T_3_/T_2_ = 0.752 ± 0.103, corresponding to a −24.78% ± 10.26% change at T_3_) and 37.5% of neurons (3 out of 8; T_3_/T_2_ = 0.702 ± 0.085, corresponding to a −29.79% ± 8.45% change at T_3_), respectively. Similarly, the Ang II-mediated inhibition of NMDA responses was not affected by AT_2_ receptor blockade. The application of the AT_2_ receptor antagonist PD123319 (5 μM) alone, throughout the measurement period, also had no effect on any of the cells tested (T_3_/T_2_ = 1.013 ± 0.039, n = 6). Notably, Ang II (1 μM) inhibited 38.4% of pyramidal cells (5 out of 13; T_3_/T_2_ = 0.588 ± 0.071, corresponding to a −41.25% ± 7.17% change at T_3_) even in the presence of the AT_2_ receptor antagonist.

### 3.2. The IRAP/AT4 Receptor for Ang IV Is Expressed in the PFC of Both Young and Adult Rats

Next, we investigated the expression of IRAP/AT4R, a candidate target of the observed inhibition, in the layers of the rat PFC in both young (the age of animals used in our electrophysiological experiments) and adult (the age more relevant to potential pathological conditions) animals. Immunohistochemical staining using anti-IRAP (D7C5) XP rabbit monoclonal antibody demonstrated the presence of IRAP/AT4R in the PFC of both 10-day-old and 6-month-old rat brains. The receptor was highly expressed in cells located in layers II–III and V–VI. The antibody stained primarily the soma of the cells, and its density was higher in the older animals (Figure 2).

### 3.3. IRAP/AT4R Is Expressed in Pyramidal Cells and GABAergic Interneurons but Not in Microglia or Astrocytes in Layer V of the Rat PFC

Staining with glutamate decarboxylase 67 (GAD67), ionized calcium-binding adapter molecule 1 (IBA1), and glial fibrillary acidic protein (GFAP) antibodies revealed the cellular localization of IRAP/AT4R in layer V of the PFC in 10-day-old and 6-month-old rat brains. IRAP/AT4R was prominently expressed in cells with a pyramidal morphology in both age groups (Figure 3A,B, examples of pyramidal cells are indicated by green arrows). Its expression was also confirmed in GABAergic interneurons, identified by anti-GAD67 staining (Figure 3A,B, upper panels, examples of GABAergic interneurons are indicated by yellow arrows), in both young and adult rats. However, the staining intensity was stronger in adult rats. IRAP/AT4R was not expressed in astrocytes, identified by anti-GFAP staining (Figure 3A,B, lower panels), or in microglia, identified by anti-IBA1 staining (Figure 3A,B, middle panels). Examples of astrocytes and microglia in Figure 3A,B are indicated by red arrows.

### 3.4. Ang IV Inhibited NMDA-Induced Inward Currents in Layer V Pyramidal Neurons of the Rat PFC

Ang IV (1 nM, 10 nM, 100 nM, 1 μM) application, similar to Ang II (Figure 1), significantly and concentration-dependently inhibited the NMDA-induced current at T_3_ compared to that measured at T_2_ in a subset (36–43%) of pyramidal cells (1 nM, −25.22 ± 3.41%, n = 6/14; 10 nM, −31.60 ± 5.04%, n = 5/14; 100 nM, −36.71 ± 4.14%, n = 10/23; 1 μM, −50.69 ± 12.57%, n = 5/13; *p* < 0.01 for all concentrations, compared with control; Figure 4A,B). Interestingly, at the highest concentration (1 μM), potentiation was also observed in sporadic cases.

### 3.5. Synaptic Isolation of Pyramidal Neurons Did Not Abolish the Inhibition of NMDA Currents by Ang IV

When pyramidal neurons were synaptically isolated from neighboring cells by the application of a Ca^2^⁺-free solution or TTX (0.5 µM)-containing aCSF throughout the measurement, this did not interfere with the inhibitory effect of Ang IV (100 nM) on NMDA responses (−37.75 ± 7.64%, n = 9/16; −28.39 ± 4.74%, n = 8/15, respectively; *p* < 0.01 for both, compared with controls; Figure 5).

### 3.6. Ang IV-Induced Inhibition of NMDA Receptors in Layer V Pyramidal Cells of the Rat PFC Was Reproduced by an IRAP Inhibitor

Given that Ang IV is proposed to inhibit the catalytic activity of IRAP via the AT_4_ receptor, we tested the effects of the IRAP inhibitor LVVYP-H7. It reproduced the inhibitory effects of Ang IV on NMDA receptor-mediated currents (10 nM, −32.47 ± 6.33% at T_3_, n = 6, *p* < 0.01; 100 nM, −26.54 ± 6.85% at T_3_, n = 5, *p* < 0.01 in both cases; Figure 6).

## 4. Discussion

The primary finding of this study is that AT4R/IRAP is abundantly expressed in layers II–III and V–VI of the mPFC in both young (10-day-old) and adult (6-month-old) rats. Cellular localization analysis revealed that AT4R/IRAP is prominently expressed in pyramidal neurons and GABAergic interneurons within layer V, while no expression was detected in astrocytes or microglia, highlighting its selective neuronal distribution. Furthermore, functional experiments conducted in young animals demonstrated that Ang IV significantly inhibits NMDA receptor-mediated currents in a subpopulation of layer V pyramidal neurons in the PFC. This inhibitory effect is most likely mediated by the suppression of the aminopeptidase activity of AT4R/IRAP. Notably, the effect was observed across a broad concentration range of Ang IV, spanning 1 nanomolar to 1 micromolar, with intensity showing a concentration-dependent trend.

Our previous research has identified several modulatory influences on NMDA receptors within the PFC, including mechanisms mediated by dopaminergic and purinergic signaling. These modulatory influences, involving both pre- and postsynaptic processes, are typically restricted to a subset of the heterogeneous population of layer V pyramidal neurons, underscoring the complexity of regulating pyramidal cell activity within this layer [17,18,19,26,27]. Our most recent publication reported the potentiation of NMDA receptors by AT_1_ angiotensin receptor activation in layer V pyramidal neurons of the rat PFC [16]. In that study, we observed that in a small subset of pyramidal cells that did not exhibit potentiation in response to Ang II, higher (micromolar) concentrations of Ang II induced a modest inhibitory trend, characterized by a reduction in NMDA receptor function. Furthermore, this phenomenon was not affected by the AT_1_ receptor antagonist eprosartan, the AT_2_ receptor antagonist PD 123319, or the synaptic isolation of the pyramidal cells. Therefore, we hypothesized that this inhibitory effect may be mediated by the conversion of Ang II into shorter angiotensin peptides, with Ang IV playing a central role in this modulation, potentially via an as-yet unidentified mechanism.

Our present, more detailed analysis of this phenomenon confirms the presence of inhibitory responses to 1–3 µM Ang II on NMDA currents in a subpopulation of layer V pyramidal cells and demonstrates that this inhibition is unaffected by AT_1_ or AT_2_ receptor antagonism. Furthermore, our data reveal that Ang IV, a hexapeptide derivative of angiotensinogen potentially converted from Ang II by aminopeptidases A and B, also inhibits NMDA-induced ion currents in a subpopulation of layer V pyramidal cells in the rat PFC.

Radiolabeled Ang IV binding sites have been reported in the brains of rodents and other mammals, including humans, with a high degree of cross-species consistency [13,14,28,29,30,31,32,33]. These binding sites are distributed across multiple brain regions, with the highest densities observed in the neocortex. Notably, the expression pattern of IRAP, a zinc-dependent aminopeptidase, closely matches the distribution of Ang IV binding sites as determined by radioligand binding studies. In the adult rat brain, IRAP expression is highly localized to the cerebral cortex, and IRAP-immunoreactive cells have also been identified in the cerebral cortex of postmortem human brains [34,35,36]. In line with previous findings, this study demonstrates a prominent protein-level expression of AT4R/IRAP in the mPFC of Wistar rats. The expression was abundant in both layers II–III and layers V–VI in young rats (9–12 days old) as well as in adult rats (6 months old).

AT4Rs were hypothesized to be restricted primarily to neurons throughout the brain [8]. Supporting this, IRAP expression was found to be exclusively localized to neurons in the neocortex [34,36]. Consistent with the consensus in the scientific literature, this study identified AT4R/IRAP expression in neurons exhibiting pyramidal morphology, as well as in GABAergic interneurons within layer V of the rat PFC. AT4Rs have also been suggested to reside on astrocytes, although limited evidence supports this hypothesis [8]. IRAP expression has been observed in astrocytes within rat astroglial cell cultures and in activated astrocytes surrounding dense amyloid beta deposits in Alzheimer’s disease (AD) mouse brains [34,37,38]. However, in this study, we did not detect AT4R/IRAP expression at the protein level in astrocytes or microglia, either in young or adult rats, within the PFC.

Our electrophysiological analysis revealed a novel modulatory role of Ang IV on NMDA receptor function in the rat PFC. Unfortunately, no commercially available antagonist for the AT_4_ receptor was accessible during our project, limiting our ability to selectively antagonize the receptor and confirm the AT_4_ receptor specificity of the Ang IV-mediated inhibition. Nevertheless, it is well-established that Ang IV, through AT4R, inhibits the aminopeptidase activity of IRAP. The observed concentration-dependent inhibition, along with a similar effect induced by the specific IRAP inhibitor LVVYP-H7, strongly supports the hypothesis that the inhibitory effect is mediated via the AT4R/IRAP signaling pathway. Moreover, the Ang IV-mediated inhibition was not abolished by exocytosis inhibition using Ca^2^⁺-free aCSF or by blocking Na⁺-dependent action potentials with tetrodotoxin, suggesting that the effect is not mediated by interneurons but rather directly affects pyramidal cells in layer V of the PFC, which our histochemical analysis confirmed to abundantly express AT4Rs.

Although various signal transduction pathways for Ang IV have been proposed over the years, a complete consensus has yet to be reached. As mentioned previously, AT4R has been identified as analogous to the transmembrane enzyme IRAP, with Ang IV acting as a potent competitive inhibitor of IRAP [10,11]. However, alternative studies have suggested that the hepatocyte growth factor receptor, c-Met, may also serve as a primary target for Ang IV [39]. In this study, the IRAP inhibitor LVVYP-H7 produced results comparable to those of Ang IV, supporting the hypothesis that the blockade of IRAP mediates the Ang IV-induced inhibition of NMDA currents in the rat PFC observed in our experiments. The AT4R/IRAP pathway may influence NMDA receptor function through several mechanisms, one of which involves the modulation of the kinase and phosphatase activities that regulate NMDA receptor function in various brain regions [40]. AT4R has been reported to stimulate several downstream phosphorylation mechanisms, including the phosphatidylinositol 3-kinase (PI3K) pathway and protein kinase B (Akt) activation [41], as well as upregulating the activity of protein phosphatases [42]. Alternative possibilities may involve the Ang IV-mediated modulation of nitric oxide levels, which could induce the blockade of NMDA receptors [43,44]. Notably, c-Met activation has been reported to influence the expression levels of NMDA receptor subunits in the mouse hippocampus [45]. Further research should address the downstream mechanisms underlying the Ang IV-mediated inhibition of NMDA currents in the present work.

It is not unusual to observe that only a specific subset of layer V pyramidal neurons participates in certain interactions, as pyramidal neurons in the PFC show considerable heterogeneity in their morphological and physiological characteristics [46]. For instance, Yang et al. classified four distinct types of pyramidal cells in layers V and VI of the rat PFC, each with unique electrophysiological profiles [47]. A similar phenomenon has been noted in our previous studies, including those investigating the Ang II-mediated potentiation of NMDA currents in the rat PFC [16,19]. Consistent with the approach employed in the latter study, we did not categorize the investigated neurons in the present study based on their morphological or electrophysiological properties. Consequently, our data do not permit the identification of specific subtypes of layer V pyramidal neurons involved in the inhibitory action. Nonetheless, our findings align with observations reported in the literature and suggest that the observed effects are not diffuse actions broadly affecting most cells within the layer. Instead, they involve specific mechanisms that modulate the activity of particular neuronal populations. However, further investigations are needed to identify the specific pyramidal cell populations involved in Ang IV-induced inhibition and to elucidate their characteristic properties.

Although Ang IV primarily and selectively binds to AT4R, it has been shown to weakly bind and activate AT_1_ receptors at very high concentrations [8]. In light of our recent publication demonstrating the AT_1_ receptor-mediated potentiation of NMDA currents in layer V pyramidal cells of the PFC [16], this may help explain the unexpected observation of sporadic potentiation in response to micromolar concentrations of Ang IV (the highest concentration applied in our experiments).

The interaction of Ang IV with various neuromodulatory transmitters and receptors has been extensively studied in brain regions associated with cognitive functions. Ang IV has been shown to potentiate depolarization-induced acetylcholine release from rat hippocampal slices [48] and to modulate hippocampal acetylcholine levels in vivo [49]. Notably, in behavioral studies, spatial memory deficits induced by nicotinic acetylcholine receptor (nAChR) antagonists were reversed by Norleual, an Ang IV analog peptide [50]. Furthermore, behavioral studies demonstrated that the activation of the nicotinic receptor system could counteract the effects of AT4R blockade, which otherwise caused severe performance impairments in the circular water maze [51]. Ang IV has been shown to induce a significant increase in extracellular dopamine concentrations in the striatum of freely moving rats [52]. The positive cognitive effects of Ang IV and des-Phe6-Ang IV on learning tasks, such as conditioned avoidance responses, recall of a passive avoidance task, and object recognition, were abolished by D1 and D2 receptor antagonists [53,54], a D3 dopamine receptor partial agonist [55], and a D4 receptor antagonist [56]. Importantly, no dedicated studies have specifically investigated the interactions between Ang IV and the cholinergic or dopaminergic systems within the PFC. Similarly, the interaction between Ang IV, glutamate, or its NMDA receptor remains insufficiently investigated, with no relevant data specific to the PFC reported in the scientific literature. Notably, hippocampal NMDA receptor expression levels were significantly elevated in streptozotocin-induced diabetic rats treated with Ang IV [57]. In contrast, an NMDA receptor antagonist had a minimal effect on AT4R-dependent long-term potentiation (LTP) in the rat hippocampus [58].

The excitatory neurotransmitter glutamate and its ionotropic NMDA receptor play a crucial physiological role in learning and memory processes [6]. However, excessive activation of the NMDA receptor can lead to excitotoxicity and apoptosis, contributing to the pathogenesis of neurodegenerative diseases [7]. NMDA receptor antagonists, such as memantine (used in the treatment of moderate to severe Alzheimer’s disease) and amantadine (used in Parkinson’s syndrome), are employed to mitigate these effects. Unlike direct NMDA receptor antagonists, our data suggest that Ang IV may function as an indirect inhibitory modulator of NMDA receptor activity. This mechanism results in a subtle modulation rather than a complete blockade, potentially allowing for a more selective influence on NMDA receptors specifically involved in cognitive functions. Such a targeted modulation may reduce the risk of severe side effects associated with direct and robust antagonism. 

In addition to the aforementioned behavioral tests highlighting the interactions of Ang IV with cholinergic and dopaminergic neurotransmission, numerous studies have demonstrated that Ang IV and its analogs may exert beneficial effects in mitigating memory deficits in various animal models [57,59,60,61,62,63,64,65]. Additionally, IRAP has been identified as a potential pharmacological target for the modulation of cognitive functions [66,67,68]. Due to the inherently poor pharmacokinetic properties of Ang IV, significant efforts have been directed towards the identification and development of analogs through screening in virtual and chemical libraries, as well as through rational design approaches. High-affinity ligands for AT4R have been identified, alongside potent IRAP inhibitors exhibiting enhanced metabolic stability [15,69,70,71,72,73,74,75,76,77,78,79,80,81]. However, overcoming absorption, metabolism, and pharmacokinetic challenges remains crucial for developing orally bioavailable drug candidates [82]. The Ang IV-AT4R/IRAP-mediated inhibition of NMDA receptor currents in a subpopulation of layer V pyramidal neurons within the PFC, as elucidated in this study, provides additional in vitro evidence supporting the therapeutic potential of Ang IV-IRAP-targeting pharmacological strategies. This finding, in conjunction with our recent publication [16], further extends the understanding of Ang-mediated effects on neurotransmitters and the brain structures relevant to higher-order cognitive functions, highlighting the modulation of the glutamatergic system, which is particularly important for cognition.

The behavioral pharmacological experiments referenced in the previous section, demonstrating the beneficial effects of Ang IV and its analogs, have typically been conducted in young adult rodents. Although our electrophysiological analysis was performed using brain slices from young rats, which may limit the direct applicability of these findings to older age groups, our histochemical data suggest that the expression of AT4Rs in the PFC of adult rats is comparable to that observed in younger animals. This finding underscores the relevance of our results in elucidating the pathophysiology of cognitive disorders associated with aging. Further studies are required to fully comprehend the implications of our findings for age-related cognitive decline. In vitro histochemical and electrophysiological analyses using brain slices from both adult and aged rats, as well as behavioral assays in older animals, are necessary to address this issue. Our study, however, contributes to the understanding of the cellular mechanisms underlying the therapeutic potential of Ang IV-mediated interventions, thereby encouraging further investigation in this area. Specifically, the activation of the Ang IV receptor appears to negatively modulate excitatory glutamatergic signaling, which may mitigate NMDA receptor-mediated excitotoxicity and neurodegeneration. Furthermore, we present additional evidence supporting AT4R/IRAP as a promising pharmacological target for pathological neuropsychiatric conditions.

## Figures and Tables

**Figure 1 biomedicines-13-00071-f001:**
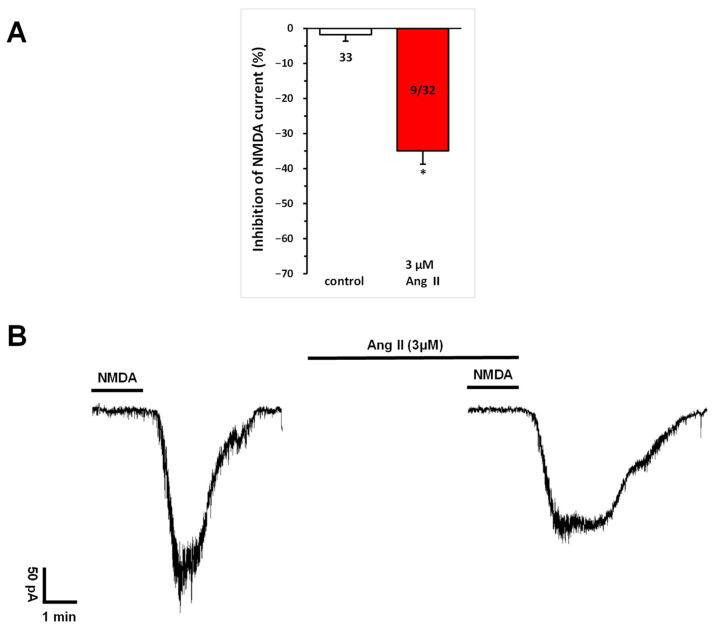
Inhibitory effect of Ang II on NMDA-induced currents in layer V pyramidal neurons of the medial prefrontal cortex in 9–12-day-old rats. Whole-cell patch-clamp recordings were performed at a holding potential of −70 mV. NMDA (30 µM) in aCSF was applied three times (T_1_, T_2_, T_3_) for 1.5 min, with 10 min intervals between applications. Ang II (3 μM) was superfused for 5 min before and during T_3_. (**A**) Inhibitory effects of Ang IV on the NMDA current at T_3_ are expressed as the percentage inhibition of the response measured at T_2_. Data are presented as mean ± SEM. Red columns represent the percentage inhibition of NMDA-induced current responses (T_3_/T_2_) in cells where Ang II inhibited NMDA receptor-mediated ion currents. The number of responsive cells out of the total number of cells tested is shown in the red columns. * *p* < 0.01, indicating a significant difference from controls. (**B**) Representative tracing of a current response to 30 μM NMDA at T_2_ and T_3_, in the presence of 3 μM Ang II superfused for 5 min before and during T_3_.

**Figure 2 biomedicines-13-00071-f002:**
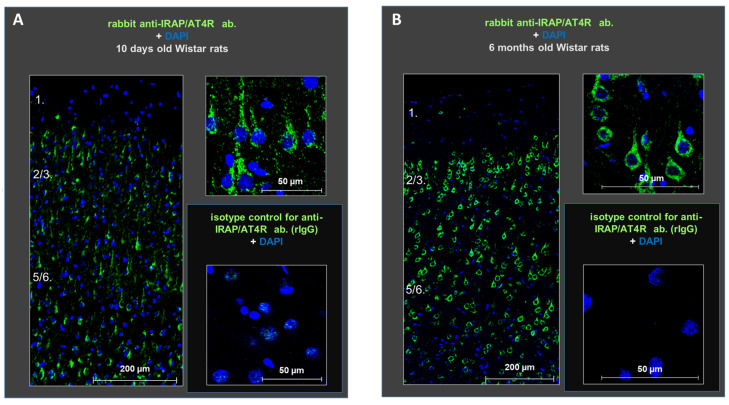
Protein expression of IRAP (green) in distinct layers of the medial prefrontal cortex in young (10-day-old) (**A**) and adult (6-month-old) (**B**) rats. The receptor is highly expressed in the cells of layers II–III and V–VI at both ages. DAPI (blue) was used as a counterstain. Scale bars: 200 μm (left panels in (**A**,**B**)) and 50 μm (right panels in (**A**,**B**)).

**Figure 3 biomedicines-13-00071-f003:**
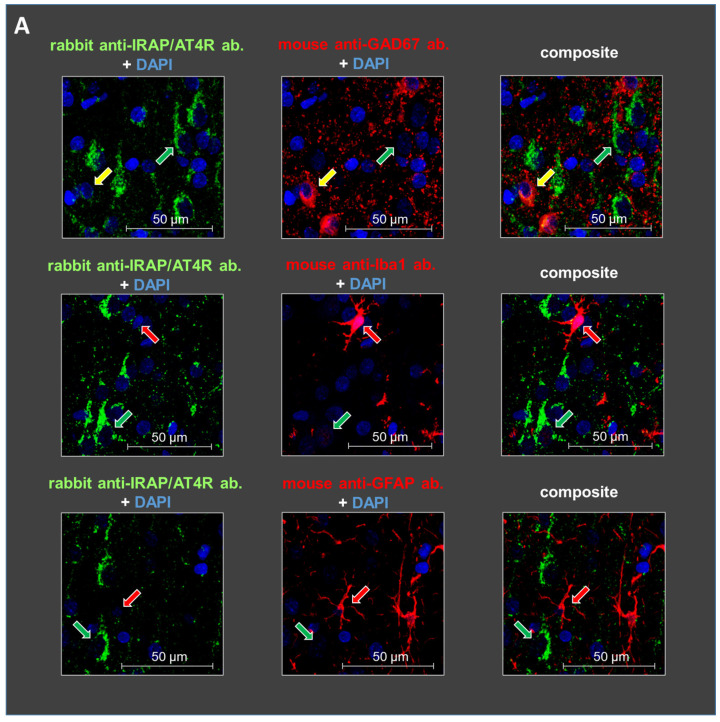
Representative images of the immunofluorescent detection of IRAP in distinct cell types in layer V of the mPFC in (**A**) young (10-day-old) and (**B**) adult (6-month-old) rats. The receptor is highly expressed in cells with a pyramidal morphology (green arrows) and in GABAergic interneurons (yellow arrows) but not in microglia or astrocytes (red arrows). GAD67, IBA1, and GFAP were used as markers. DAPI was used as a counterstain (blue). Notably, IRAP staining is weaker in 10-day-old animals compared to 6-month-old rats. Furthermore, this reduction is particularly pronounced in GABAergic interneurons but to a lesser extent in pyramidal cells in the 10-day-old rats. Scale bar: 50 μm.

**Figure 4 biomedicines-13-00071-f004:**
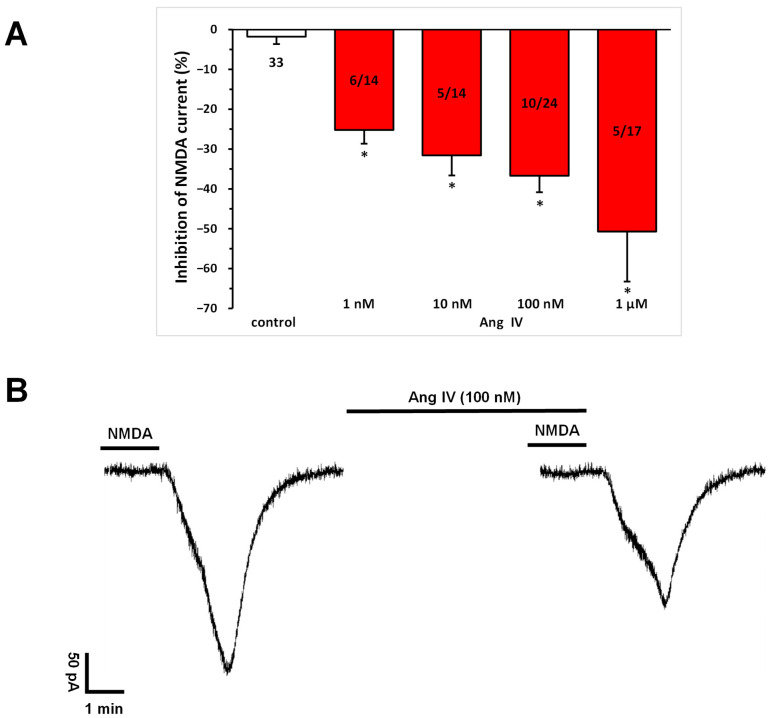
Inhibitory effects of Ang IV on the NMDA-induced current in layer V pyramidal neurons of the mPFC in young (9–12 days old) rats. Whole-cell patch-clamp recordings were performed at a holding potential of −70 mV. NMDA (30 µM) in aCSF was applied three times for 1.5 min (T_1_, T_2_, T_3_), with a 10 min interval between applications. Ang IV (1 nM to 1 μM) was superfused 5 min before and during T_3_. (**A**) Inhibitory effects of Ang IV on the NMDA current at T_3_ are expressed as the percentage inhibition of the response measured at T_2_. Data are presented as mean ± SEM. Red columns represent the percentage inhibition of NMDA-induced current responses in cells where Ang IV inhibited NMDA receptor-mediated ion currents. The number of responsive cells out of the total number of cells tested is indicated in the red columns. * *p* < 0.01; significant difference from controls. (**B**) Representative tracing of a current response to 30 μM NMDA after T_2_ and T_3_ and in the presence of Ang IV (100 nM) for 5 min before and during T_3_.

**Figure 5 biomedicines-13-00071-f005:**
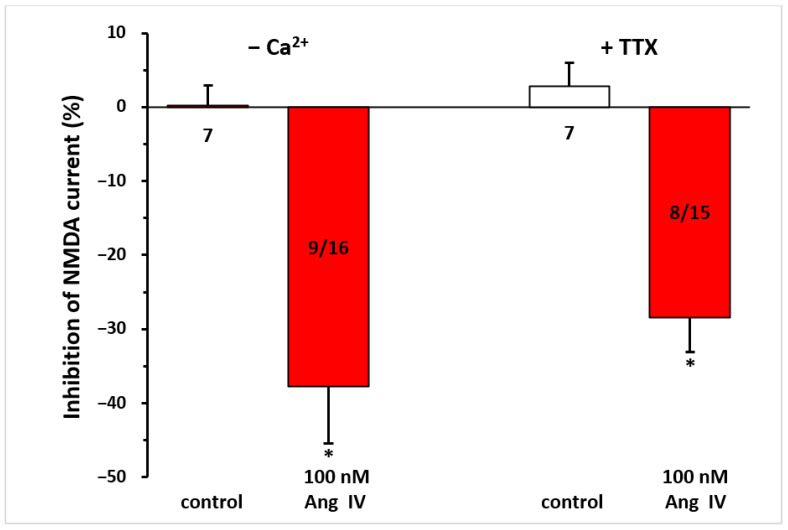
The inhibitory effect of Ang IV persisted during synaptic isolation by a Ca^2^⁺-free solution or TTX-containing aCSF on the NMDA-induced current in layer V pyramidal neurons of the mPFC in young (9–12 day-old) rats. Whole-cell patch-clamp recordings were performed at a holding potential of −70 mV. NMDA (30 µM) was applied three times for 1.5 min (T_1_, T_2_, T_3_), with a 10 min interval between applications. Ang IV (100 nM) was superfused 5 min before and during T_3_, while the Ca^2^⁺-free solution or TTX (0.5 μM) was superfused throughout the experiment. The inhibitory effects of Ang IV on the NMDA current at T_3_ are expressed as the percentage inhibition of the response measured at T_2_. Data are presented as mean ± SEM. Red columns represent the percentage inhibition of NMDA-induced current responses in cells where Ang IV inhibited NMDA receptor-mediated ion currents. The number of responsive cells out of the total number of cells tested is indicated in the red columns. * *p* < 0.01; significant difference from controls.

**Figure 6 biomedicines-13-00071-f006:**
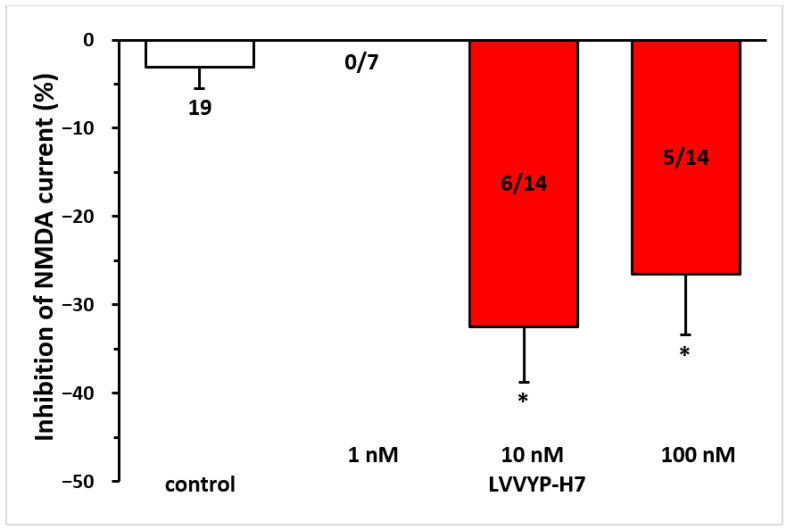
The inhibitory effect of Ang IV was reproduced by the IRAP inhibitor LVVYP-H7 on the NMDA-induced current in layer V pyramidal neurons of the mPFC in young (9–12 days old) rats. Whole-cell patch-clamp recordings were performed at a holding potential of −70 mV. NMDA (30 µM) in aCSF was applied three times for 1.5 min (T_1_, T_2_, T_3_), with a 10 min interval between applications. LVVYP-H7 (1 nM–100 nM) was superfused for 5 min before and during T_3_. The inhibitory effects of LVVYP-H7 on the NMDA current at T_3_ are expressed as the percentage inhibition of the response measured at T2. Data are presented as mean ± SEM. Red columns represent the percentage inhibition of NMDA-induced current responses in cells where LVVYP-H7 inhibited NMDA receptor-mediated ion currents. The number of responsive cells out of the total number of cells tested is indicated in the red columns. * *p* < 0.01, indicating a significant difference from the control.

## Data Availability

The data that support the findings of this study are available from the corresponding author upon reasonable request.

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
