# Peer review of "Angiotensin IV Receptors in the Rat Prefrontal Cortex: Neuronal Expression and NMDA Inhibition"

_biomedicines, 2024, doi:10.3390/biomedicines13010071_

Round 1
Reviewer 1 Report
Comments and Suggestions for Authors
This manuscript titled explores the modulatory effects of Angiotensin IV (Ang IV) on NMDA receptor function in the prefrontal cortex (PFC), a critical region for cognitive processes and neuropsychiatric disorders. The study is well-conceived and employs a robust methodology, including patch-clamp electrophysiology and immunohistochemistry, to elucidate the cellular distribution and functional significance of AT4R/IRAP in both young and adult rats. The findings are particularly novel, suggesting that Ang IV-mediated inhibition of NMDA receptor currents could represent a potential therapeutic target for cognitive impairments and neurodegenerative conditions. There are certain questions that need to be carefully addressed:
1. How do the inhibitory effects of Ang IV vary across different morphological subtypes of layer V pyramidal neurons?
2. What downstream signaling pathways (e.g., CaMKII, c-Met) are activated by Ang IV-AT4R/IRAP interactions, and how do they mediate NMDA receptor inhibition?
3. Can the effects of Ang IV on NMDA receptor-mediated currents be replicated in adult rats, and what are the implications for age-related cognitive decline?
4. Are pharmacokinetically optimized Ang IV analogs capable of eliciting similar effects in vivo, and how do they compare to Ang IV itself in terms of therapeutic potential?
5. What interactions exist between Ang IV and other neuromodulatory systems, such as dopaminergic or cholinergic pathways, in the PFC?
Author Response
Thank you very much for your thorough and supportive review.
In response to your questions, suggestions, and concerns regarding the manuscript, we have made the following revisions. All changes are highlighted in blue text within our responses and in the manuscript.
„How do the inhibitory effects of Ang IV vary across different morphological subtypes of layer V pyramidal neurons?”
We have expanded the discussion regarding the variability of the inhibitory effects of Ang IV across different morphological subtypes of layer V pyramidal neurons, highlighting that further experiments are needed to address this issue directly. („Consistent with the approach employed in the latter study”)
„What downstream signaling pathways (e.g., CaMKII, c-Met) are activated by Ang IV-AT4R/IRAP interactions, and how do they mediate NMDA receptor inhibition?”
We have expanded the discussion to include a description of the possible downstream signaling pathways involved in this interaction supported by additional references. („The AT4R-IRAP pathway may influence NMDA receptor function through several mechanisms…”)
„Can the effects of Ang IV on NMDA receptor-mediated currents be replicated in adult rats, and what are the implications for age-related cognitive decline?”
We have expanded the discussion to address the potential age-related consequences of our findings. („The behavioral pharmacological experiments, referenced in the previous section, demonstrating the beneficial effects of Ang IV and its analogs, have typically been conducted in young adult rodents. …”)
„Are pharmacokinetically optimized Ang IV analogs capable of eliciting similar effects in vivo, and how do they compare to Ang IV itself in terms of therapeutic potential?”
We have expanded the discussion to include information on pharmacokinetically optimized Ang IV analogs and their beneficial effects on cognitive functions, supported by additional relevant references. („In addition to the aforementioned behavioral tests highlighting the interactions of Ang IV with cholinergic and dopaminergic neurotransmission, numerous studies have demonstrated that…”)
„What interactions exist between Ang IV and other neuromodulatory systems, such as dopaminergic or cholinergic pathways, in the PFC?”
We have expanded the discussion to address this question, noting that no dedicated studies have specifically investigated the interactions between Ang IV and the dopaminergic or cholinergic systems in the PFC. („Importantly, no dedicated studies have specifically investigated…”)

Reviewer 2 Report
Comments and Suggestions for Authors
Angiotensin IV Receptors in the Rat Prefrontal Cortex: Neu-ronal Expression and NMDA Inhibition
Review
This study elucidated the Ang IV-AT4R/IRAP-mediated inhibition of NMDA receptor currents in a subpopulation of layer V pyramidal neurons within the PFC. While our electrophysiological analysis was conducted using brain slices from young rats, which may limit the direct applicability of these findings to older age groups, our current histochemical data indicate that the expression of AT4Rs in the PFC of adult rats is similar to that in younger animals. This finding underscores the relevance of our results in elucidating the pathophysiology of cognitive disorders associated with ageing.
Although well written, the MS needs several improvements in linguistic and grammar aspects.
Abstract (edited by the reviewer as an example; there are minor issues, but valuable)
Background: NMDA-type glutamate receptors are fundamental to neuronal physiology and pathophysiology. The prefrontal cortex (PFC), a key region for cognitive function, is heavily implicated in neuropsychiatric disorders, positioning the modulation of its glutamatergic neuro-transmission as a promising therapeutic target. Our recently published findings indicate that AT1 receptor activation enhances NMDA receptor activity in layer V pyramidal neurons of the rat PFC. At the same time, it suggests that alternative angiotensin pathways, presumably involving AT4 angiotensin receptors (AT4R), might exert inhibitory effects. Angiotensin IV (Ang IV) and its analogues have demonstrated cognitive benefits in animal models of learning and memory deficits. Methods: Immunohistochemistry and whole-cell patch-clamp techniques were used to map the cell-type-specific localization of AT4R, identical to insulin-regulated aminopeptidase (IRAP), and to investigate the modulatory effects of Ang IV on NMDA receptor function in layer V pyramidal cells of the rat PFC. Results: AT4R/IRAP expression was detected in pyramidal cells and GABAergic interneurons, but not in microglia or astrocytes, in layer V of the PFC in 9-12-day-old and 6-month-old rats. NMDA (30 μM) induced stable inward cation currents, significantly inhibited by Ang IV (1 nM - 1 μM) in a subset of pyramidal neurons. This inhibition was reproduced by the IRAP inhibitor LVVYP-H7 (10-100 nM). Synaptic isolation of pyramidal neurons did not affect the Ang IV-mediated inhibition of NMDA currents. Conclusion: Ang IV/IRAP-mediated inhibition of NMDA currents in layer V pyramidal neurons of the PFC may represent a potential pharmacological target for cognitive impairments and related neuropsychiatric disorders.
Introduction
Put a dot after the references (in the whole MS...).
f.e. dementias (1,2).
M and M
Please provide all the details of your protocol and methods. Revise this part.
References
Please check the citation style and revise. There are a lot of irregularities.
Images / The quality is good; please try to replace some of the pictures if you have better-quality photos.
Comments on the Quality of English LanguageEnglish should be improved.
Author Response
Thank you very much for your thorough and supportive review.
In response to your questions, suggestions, and concerns regarding the manuscript, we have made the following revisions. All changes are highlighted in blue text within our responses and in the manuscript.
„Although well written, the MS needs several improvements in linguistic and grammar aspects.
Abstract (edited by the reviewer as an example; there are minor issues, but valuable)”
We are grateful for the thorough grammatical review and your valuable suggestions. The abstract has been revised, and your recommendations have been incorporated.
„Introduction. Put a dot after the references (in the whole MS...).”
We have checked the placement of the dots after the references throughout the manuscript and have made the necessary corrections.
„Please provide all the details of your protocol and methods.”
We have included the additional necessary details in the Materials and Methods section.
„Please check the citation style and revise. There are a lot of irregularities.”
With the help of the EndNote program, we have revised the references, and the appropriate format was generated by the program.
„Images / The quality is good; please try to replace some of the pictures if you have better-quality photos.”
The reduced quality of the images was due to a technical issue that occurred during the upload or editing process. We have replaced the images with the original, higher-quality versions.
„Comments on the Quality of English Language: English should be improved.”
With the assistance of a native English speaker, we have thoroughly revised the manuscript to improve the clarity and quality of the language.
